# Lactide and Ethylene Brassylate-Based Thermoplastic Elastomers and Their Nanocomposites with Carbon Nanotubes: Synthesis, Mechanical Properties and Interaction with Astrocytes

**DOI:** 10.3390/polym14214656

**Published:** 2022-11-01

**Authors:** Carlos Bello-Álvarez, Agustin Etxeberria, Yurena Polo, Jose-Ramon Sarasua, Ester Zuza, Aitor Larrañaga

**Affiliations:** 1Department of Mining-Metallurgy Engineering and Materials Science, POLYMAT, Faculty of Engineering in Bilbao, University of the Basque Country (UPV/EHU), Plaza Ingeniero Torres Quevedo 1, 48013 Bilbao, Spain; 2Advanced Polymers and Materials: Physics, Chemistry and Technology Department, POLYMAT, University of the Basque Country (UPV/EHU), 20018 Donostia-San Sebastián, Spain; 3Polimerbio SL, 20014 Donostia-San Sebastian, Spain

**Keywords:** ethylene brassylate, lactide, bioresorbable polymer, elastomer, neural regeneration

## Abstract

Polylactide (PLA) is among the most commonly used polymers for biomedical applications thanks to its biodegradability and cytocompatibility. However, its inherent stiffness and brittleness are clearly inappropriate for the regeneration of soft tissues (e.g., neural tissue), which demands biomaterials with soft and elastomeric behavior capable of resembling the mechanical properties of the native tissue. In this work, both L- and D,L-lactide were copolymerized with ethylene brassylate, a macrolactone that represents a promising alternative to previously studied comonomers (e.g., caprolactone) due to its natural origin. The resulting copolymers showed an elastomeric behavior characterized by relatively low Young’s modulus, high elongation at break and high strain recovery capacity. The thermoplastic nature of the resulting copolymers allows the incorporation of nanofillers (i.e., carbon nanotubes) that further enable the modulation of their mechanical properties. Additionally, nanostructured scaffolds were easily fabricated through a thermo-pressing process with the aid of a commercially available silicon stamp, providing geometrical cues for the adhesion and elongation of cells representative of the nervous system (i.e., astrocytes). Accordingly, the lactide and ethylene brassylate-based copolymers synthesized herein represent an interesting formulation for the development of polymeric scaffolds intended to be used in the regeneration of soft tissues, thanks to their adjustable mechanical properties, thermoplastic nature and observed cytocompatibility.

## 1. Introduction

Polylactides (PLA) are well-known biocompatible and bioresorbable polyesters used in several biomedical applications as drug delivery systems, implants and cellular scaffolds for tissue engineering applications [1]. However, due to its inherent stiffness (Young’s modulus (E) of 1–3.5 GPa for poly(D,L-lactide)(PDLLA) and 2.7–4.1 GPa for poly(L-lactide)(PLLA)) and the brittleness (elongation at break (ε) of 10% for PDLLA and PLLA) [2], the use of PLA for the regeneration of many tissues in the human body is restricted, because of the elastomeric and soft behavior that these tissues generally present. Thus, several strategies such as increasing the molecular mass, cross-linking, incorporating nanoparticles/nanofibers [3] or copolymerization [4] are regularly considered to modulate the mechanical properties of PLA [5]. Among them, the copolymerization of lactide with (macro)lactones (e.g., caprolactone [6] or ethylene brassylate (EB) [4]) has been widely explored as a strategy to provide enhanced flexibility to the polymeric chain and the resulting polymeric devices [5]. EB is a macrolactone extracted from castor oil that represents a more economical approach to alternative counterparts (e.g., ε-caprolactone). In addition, its natural origin allows us to avoid dependence on fossil feedstock [7]. In the present study, EB was used for the synthesis of LA-based copolymers with adjustable mechanical properties that can match those of the nervous system, which shows Young’s modulus between 0.5 and 16 MPa and an ultimate strain of 61–81% [8,9,10]. Accordingly, lactide-ethylene brassylate (LA-EB) copolymers could represent an interesting alternative to the more widely explored poly(lactide-co-ε-caprolactone) (PLCL) copolymers [11,12].

The LA-EB copolymers synthesized in the present work show thermoplastic behavior, allowing its processing by both traditional (e.g., injection molding, extrusion) or more advanced (e.g., electrospinning, 3D printing, lithography) fabrication techniques. Consequently, complex nanostructures mimicking the extracellular matrix where cells reside can be developed, improving in this way the interaction between cells and the copolymer in terms of proliferation, adhesion and cell fate [13]. Another advantage of thermoplastic (co)polyesters is the possibility of adding (nano)particles (e.g., carbon nanotubes (CNT)) into the polymeric matrix. CNTs are one carbon atom layers rolled as a cylinder that can improve mechanical properties, provide nanoroughness [14] to the polymer surface and provide electrical conductivity to the polymeric device [15]. In the biomedical field, CNTs have shown to be an ideal substrate due to their good compatibility and the ability to promote the adhesion and proliferation of various cell lines [16], particularly for the regeneration of the nervous system [17].

In the present study, D,L-LA-co-EB (DL:EB) and L-LA-co-EB (L:EB) copolymers were synthesized by ring-opening polymerization using triphenyl bismuth as catalyst-initiator. To modulate their properties, CNTs were incorporated into the polymeric matrix at different concentrations. Finally, with the aim of studying the interaction of our materials with cells representative of the nervous system, the metabolic activity and adhesion of mouse astrocytes (C8-D1A) were assessed in both flat and nanopatterned surfaces. Astrocytes are the most abundant glial cells in the nervous system and provide support and nutrients to neurons. When an injury occurs in the nervous system, a gap can be created between the nerves. In the regeneration process to rejoin the nerve cells, astrocytes can become reactive, contributing to the formation of a glial scar, a physical and chemical barrier that prevents the correct axonal regeneration [18]. Previous studies show that polymeric materials with specific topographies can influence the astrocytes’ inflammatory response, which is translated into the bio-integration of the implanted material [19]. It has also been shown that CNTs can induce morphological changes in astrocytes, provoking even functional changes [20]. Considering the relevance of astrocytes for regenerative purposes, in the present work, their interaction with a fully implantable polymeric device is preliminarily studied after a thorough characterization of the synthesized copolymers in terms of mechanical, thermal, morphological and surface properties.

## 2. Materials and Methods

### 2.1. Materials

Ethylene brassylate (EB) monomer was supplied by Sigma Aldrich (Spain), while D,L-lactide and L-lactide monomers were provided by Corbion (The Netherlands). The triphenyl bismuth (Ph_3_Bi) catalyst was obtained from Gelest (USA). Dichloromethane and n-Hexane were supplied by Labbox (Spain). Multi-Walled Carbon Nanotubes (MWCNT), having an average diameter of 10–15 nm with 5–15 walls and 1–10 μm in length, were supplied by Arkema (France). Dulbecco’s Modified Eagle Medium (DMEM), Hanks’ Balanced Salt Solution (HBSS), Penicillin Streptomycin (P/S), Fetal Bovine Serum (FBS), AlamarBlue cell viability reagent and rhodamine-phalloidin were supplied by Fisher Scientific (Spain). Laminin, PBS (Phosphate-buffered saline), Diiodomethane, Fluoroshield with DAPI, Triton X-100 and Tween 20 were supplied by Sigma Aldrich (Spain), and 16% Formaldehyde solution was supplied by Thermo Fisher Scientific (USA). Chloroform was supplied by PanReac (Spain).

### 2.2. Synthesis

For the synthesis of the copolymers, we followed our previously described protocol [4]. Briefly, copolymers from D,L-LA or L-LA and EB were synthesized in bulk by one pot-one-step ring-opening polymerization (ROP) with a feed molar composition of 70% LA and 30% EB. Polymerizations were carried out in a round bottom flask immersed in a controlled temperature oil bath at 140 °C. When the monomers were melted, a nitrogen stream was initiated under the surface of the melt to purge the flask for 15 min. Subsequently, the catalyst (Ph_3_Bi) was added at 100:1 comonomers/catalyst molar ratio under stirring. After 72 h of reaction, the product was dissolved in dichloromethane and precipitated in excess of n-hexane (previously cooled at 4 °C) to remove the catalyst impurities and those monomers that had not reacted. Finally, the product was dried at room temperature and then subjected to a heat treatment at 100 °C for 1 h to ensure the complete elimination of any remaining solvent.

### 2.3. Incorporation of MWCNT in the Polymeric Matrix and Composite Preparation

First, the copolymer was dissolved in dichloromethane. Different amounts of MWCNT (0.1, 0.5 and 1 wt.%) were added while the polymeric solution was under stirring. This range of carbon nanotubes was selected based on our previous work [15], where the resulting polymeric matrix showed appropriate electrical conductivity and a slight stiffening with respect to the pristine polymer at these MWCNT ratios. The mixture was sonicated with a UP400St ultrasonic processor (Hielscher Ultrasonics, Germany) (with a 14 mm diameter sonotrode) for 20 min, with a 100% amplitude, in pulses of 1 s. The mixture was then transferred into a beaker under stirring until most of the solvent had been eliminated. The resultant product was vacuum-dried for 5 days to eliminate the remaining solvent. The system without MWCNT underwent the same process. Subsequently, films of 250 µm thickness were obtained by compression molding using a Collin P 200 hydraulic press (Collin, Germany). First, a predetermined amount of copolymer was melted at 140 °C for DL:EB and 160 °C for L:EB for 5 min. Then, 250 bars were applied for 45 s. To eliminate any remaining solvent, the film was placed under a vacuum and 60 °C for DL:EB and 85 °C for L:EB for 30 min. Then, the film was cut into pieces and then processed again by compression molding to obtain the final film. The films were stored at −20 °C until their mechanical characterization to avoid any possible aging during their storage. A representative figure of the obtained films after the synthesis and processing is included in the Appendix A.

To create a nanopattern on the surface of the films, lithography was carried out, as previously reported by us [11]. Briefly, a commercially available silicon stamp with a 300 nm period nanopattern was replicated on the surface of the film by applying pressure and increasing the temperature to 200 °C [21].

### 2.4. Differential Scanning Calorimetry (DSC)

The thermal properties of the synthesized copolymers and final films were analyzed by Differential Scanning Calorimetry (DSC) on a Q200 model (TA Instruments, USA). Samples weighing between 5 and 10 mg were encapsulated in hermetic aluminum pans to study the glass transition temperatures (T_g_), melting temperature (T_m_) and melting enthalpy (ΔH_m_). Two scans were performed. Both scans were carried out from −80 to 180 °C in the case of DL:EB and from −80 to 200 °C in the case of L:EB, at 20 °C min^−1^.

### 2.5. Gel Permeation Chromatography (GPC)

The molecular weight and dispersity index of the synthesized copolymers and final films were determined in a Waters 1515 GPC device equipped with two Styragel columns (102–104 Ǻ) (Waters, Milford, MA, USA). Samples were prepared at a concentration of 10 mg mL^−1^ in chloroform and filtered with a 0.2 μm pore size filter.

### 2.6. Nuclear Magnetic Resonance (NMR)

Proton and carbon nuclear magnetic resonance (^1^H NMR and ^13^C NMR) spectra were acquired in a Bruker Avance DPX 300 (Bruker, USA) at 300.16 MHz and 75.5 MHz of resonance frequency, respectively, using 5 mm O.D. sample tubes. From solutions of 0.7 mL of deuterated chloroform, all spectra were obtained at room temperature. Experimental conditions are the same as reported in the bibliography [4]. For ^1^H NMR: 10 mg of sample; 3 s acquisition time; 1 s delay time; 8.5 µs pulse; spectral width 5000 Hz and 32 scans. For ^13^C NMR: 40 mg, inverse gated decoupled sequence; 3 s acquisition time; 4 s delay time; 5.5 µs pulse; spectral width 18,800 Hz and more than 10,000 scans.

Using the tables of structural determination from Prestch et al. [22], the different signals were assigned. By averaging the values of molar contents and the LA-EB dyad relative molar fractions that were obtained by means of ^1^H and ^13^C NMR spectroscopy, the copolymer composition data, average sequence lengths, and randomness character were calculated. The number average sequence lengths (*l*_i_), the Bernoullian random number average sequence lengths (*l*_i_)_random_ and the randomness character (*R*) were obtained using equations (Equations (1)–(4)):(1)$$l_{LA}=\frac{\left(LA-LA\right)+\frac{1}{2}\left(LA-EB\right)}{\frac{1}{2}\left(LA-EB\right)}=\frac{2\left(LA\right)}{\left(LA-EB\right)}\;\;$$
(2)$$l_{EB}=\frac{\left(EB-EB\right)+\frac{1}{2}\left(LA-EB\right)}{\frac{1}{2}\left(LA-EB\right)}=\frac{2\left(EB\right)}{\left(LA-EB\right)}\;\;$$
(3)$${\left(l_{LA}\right)}_{random}=\frac{1}{\left(EB\right)}\;;{\left(l_{EB}\right)}_{random}=\frac{1}{\left(LA\right)}\;$$
(4)$$R=\frac{{\left(l_{LA}\right)}_{random}}{l_{LA}}=\frac{{\left(l_{EB}\right)}_{random}}{l_{EB}}\;\;$$
where (*LA*) and (*EB*) are the lactide and ethylene brassylate molar fractions, and (*LA* − *EB*), (*LA* − *LA*) and (*EB* − *EB*) are the *LA* − *EB*, *LA* − *LA* and *EB − EB* average dyad relative molar fractions, respectively.

### 2.7. Electron Microscopy Analysis

Scanning electron microscopy (SEM) (HITACHI S-4800, Japan) was used to analyze the surface morphology of the scaffolds. Samples were coated with a 150 Å layer of gold in a JEL Ion Sputter JFC-1100 at 1200 V and 5 mA before being analyzed.

The dispersion quality of CNTs on the polymeric matrix was studied by Transmission electron microscopy (TEM) (JEOL 1400 Plus) (JEOL, Tokyo, Japan).

### 2.8. Tensile Test

The mechanical properties of the resulting films were determined by a tensile test with an Instron 5565 testing machine (Instron, USA) at a crosshead displacement rate of 10 mm min^−1^ at room temperature (21 ± 2 °C) and human body temperature (37 ± 2 °C). Samples of 100 mm in length and 10 mm wide were cut from films of 250 µm average thickness. The initial distance between the clamps was fixed at 50 mm. All properties were determined as the mean value of at least four determinations.

### 2.9. Contact Angle

Static contact angle measurements were carried out at room temperature by sessile drop method in order to evaluate the surface energy of copolymeric films. To calculate it, the Fowkes’ method was applied by equation (Equation (5)), where $\gamma_{l}=\gamma_{l}^{d}+\gamma_{l}^{p}$ are the total, $\gamma_{l}^{d}$ are the dispersive and $\gamma_{l}^{p}$ are the polar components of the liquid surface energy. Water ($\gamma_{l}^{d}\;$= 21.8 mJ/m^2^ and $\gamma_{l}^{p}\;$= 51 mJ/m^2^) and diiodomethane ($\gamma_{l}^{d}\;$= 50.8 mJ/m^2^ and $\gamma_{l}^{p}$= 0 mJ/m^2^) were used as the test liquids [23]. Briefly, at least 4 drops of 10 µL were placed on the surface of the films. Then, average values of contact angles were obtained using a Krüss Drop Shape Analyzer DSA100.
(5)$$\gamma_{l}\left(1+cos\theta \right)=2\left(\sqrt{\gamma_{l}^{d}\gamma_{s}^{d}}+\sqrt{\gamma_{l}^{p}\gamma_{s}^{p}}\right)\;\;$$
where $\gamma_{s}^{d}$ and $\gamma_{s}^{p}$ are the dispersive and polar components of the surface energy for solid surfaces, respectively.

### 2.10. Cell Viability Assay

C8-D1A cells (ATCC, USA) were cultured in Dulbecco’s Modified Eagle Medium (DMEM) supplemented with 10% FBS and 1% P/S. Cultures were maintained in a humidified atmosphere (5% CO_2_, 95% relative humidity) at 37 °C. Three circular samples of 10 mm diameter were punched out from each system and placed in a 24-well plate. The circular samples were extensively washed with ethanol and subsequently washed with PBS before being air-dried inside the biosafety cabinet. Finally, they were placed under ultraviolet light for 15 min. To enhance cell attachment, laminin (1:200) was used to cover the films and wells. Samples were incubated for 2 h at 37 °C and subsequently washed with PBS to remove the excess laminin before astrocytes were seeded. To assess the cell viability, 10,000 cells were seeded per well. After either 24 h or 48 h, the cell medium was replaced by a cell medium containing 10% of AlamarBlue reagent. Cells were incubated for 4 h, sheltered from light. Finally, the metabolic activity was determined by measuring the fluorescence (λ_ex_ = 480 nm/λ_em_ = 595 nm) of the media on a BioTek Synergy H1 plate reader (BioTek, USA).

### 2.11. Immunostaining

C8-D1A cells were seeded on the films in the same way as explained for the cell viability assay. After 48 h, cells were fixed with 4% PFA for 10 min at room temperature. Then, PFA was aspirated, and the samples were washed with sterile HBSS twice. A solution of 0.5% Triton X-100 in PBS was added to permeabilize the membrane of the cells and kept under slight agitation for 10 min. The solution was removed, and the samples were washed with PBS under slight agitation for 5 min, twice. Then, 300 µL per well of rhodamine-phalloidin (Rd/Ph) solution was added, and the plate was incubated under slight agitation for 15 min. Rd/Ph solution was removed, and the samples were washed twice with 0.1% Tween 20 in PBS for 5 min under slight agitation. Finally, samples were washed with PBS for 5 min under slight agitation and were kept in fresh PBS before observing them under a fluorescence microscope. Nuclei staining with DAPI was carried out directly on the samples by adding a drop of mounting medium before observing them under the microscope (Nikon Eclipse Ts2, Nikon, Tokyo, Japan).

### 2.12. Statistical Analysis

At least 4 measurements for AlamarBlue were assayed. Results were presented as the mean average ± SD or SEM. One-way ANOVA, non-parametric Holm–Sidak or Kruskal–Wallis tests were used to compare data from several groups, and they were followed by a Dunn’s post hoc analysis. We considered statistical significance to be * *p* < 0.05.

## 3. Results and Discussion

### 3.1. Synthesis and Initial Characterization of the Copolymers

Table 1 summarizes the initial characterization of the synthesized copolymers. The reaction was carried out at 140 °C for 3 days. In the case of DL:EB, the resulting copolymer presented a M_w_ of 242,850 g/mol and a T_g_ of 33.5 ± 2.0 °C, which are higher than the ones referenced in the bibliography for copolymers with similar composition [4]. In the case of L:EB, the copolymer had a M_w_ of 325,790 g/mol and a T_g_ of 41.0 ± 4.5 °C. Additionally, the L:EB copolymer presented a melting point with a T_m_ of 158.0 ± 1.5 °C and a crystallinity degree (χ) of 29.1% (Appendix A).

As reported in the bibliography, the conversion of LA is significantly higher than EB [4]. After the synthesis, only 13.2% and 19.4% (in weight) of EB became part of DL:EB and L:EB copolymers, respectively. As a result of this slower reaction rate, LA was consumed more quickly than EB. Along with this, the randomness character (*R*) value was 0.83 for DL:EB and 0.78 for L:EB, which led to the random distribution of sequences (*R*→1). The incorporation of EB promotes the decrease in the average sequence lengths of LA (*l**_LA_*), as has been reported in the bibliography [4]. For example, a (co)polymer of D,L-LA:EB with 97% in D,L-LA presents a *l**_LA_* of 30.81, in contrast to our DL:EB that showed an *l**_LA_* of 16.00 and of 11.36 for the L:EB counterpart. In that same study, the reported EB-unit average sequence length presents a *l**_EB_* of 0.93, which is lower than the one observed in our DL:EB (*l**_EB_* of 1.30) and L:EB (*l**_EB_* of 1.45) copolymers. As a result, the chain microstructure of our (co)polymers is more randomly distributed.

The molar compositions depicted in Table 1 were calculated by averaging the results obtained from ^1^H and ^13^C NMR spectroscopy (Appendix A).

### 3.2. Incorporation of MWCNTs in the Polymeric Matrix and Preparation of Composites

The processing conditions followed to incorporate the CNTs in the polymer matrix induced a decrease in the M_w_. Table 2 shows that DL:EB films had a final M_w_ between 190,150 g/mol and 215,390 g/mol, which represents a loss in M_w_ between 11.3% and 21.7% with respect to the initial molecular weight of the synthesized copolymer. In the case of L:EB films, with final M_w_ ranging from 210,170 g/mol to 256,410 g/mol, they experimented a decrease between 21.3% and 33.6%. In both cases, the addition of CNTs did not seem to play a major role in the observed molecular weight decrease.

Table 3 shows the thermal properties of the resulting films, which are also represented in Appendix A. For DL:EB, the T_g_ decreased from 33.5 ± 2.0 °C to values between 25.0 ± 1.5 °C and 28.0 ± 2.0 °C in the 1st scan. For L:EB, the T_g_ decreased from 41.0 ± 4.5 °C to values between 30.0 ± 1.0 °C and 37.4 ± 4.2 °C. The incorporation of CNTs seemed to slightly increase the observed T_g_ in the case of L:EB. This might be ascribed to the capacity of CNTs to reduce the mobility of the polymeric chains in the amorphous phase [24]. It should be noticed that the DSC thermograms did not show any enthalpy relaxation around the T_g_ in the first scan, which confirms that the samples did not undergo any aging process that may affect their final mechanical properties.

The successful incorporation of MWCNTs and their dispersion in the polymeric matrix were assessed by transmission electron microscopy (TEM). Figure 1 shows the TEM micrographs of DL:EB films with either 0.5% (Figure 1a) or 1% (Figure 1c) CNTs, while Figure 1b,d shows L:EB with 0.5% and 1% CNT, respectively. The incorporation of 1% CNTs resulted in the observation of many bundles of CNTs. In contrast, when 0.5% CNTs were incorporated, a relatively good dispersion with fewer bundles was observed.

### 3.3. Mechanical Properties

Stress–strain curves of the synthesized copolymers at room temperature are presented in Figure 2. The DL:EB samples showed a typical elastomeric behavior with relatively low Young’s modulus and high elongation at break. In contrast, the L:EB samples showed a plastic behavior, characterized by a well-defined yield point and lower elongation at break and strain recovery after the break.

Figure 3 shows the elastomeric behavior of both (co)polymers DL:EB and L:EB at body temperature (37 °C). L:EB samples showed the strain-hardening phenomena at high deformations, whereas DL:EB samples displayed a more linear behavior, keeping a constant engineering stress value with the deformation.

The mechanical properties of DL:EB and L:EB copolymers are also summarized in Table 4 and Appendix A. The incorporation of D,L- or L-lactide in the copolymer had a strong impact on the strength-related mechanical properties. In this sense, those copolymers synthesized with L-lactide showed much higher values for Young’s modulus and strength but lower values of elongation at break and strain recovery after break. As an illustration, the films with no CNTs had Young’s modulus and yield strength at room temperature of 2076.6 ± 171.0 and 14.1 ± 2.5 MPa, respectively, for L:EB films, whereas these values were much lower for the DL:EB counterpart (Young’s modulus: 390.3 ± 28.8 MPa/Yield strength: 0.7 ± 0.1). However, the elongation at break and strain recovery after break were respectively 177.2 ± 7.6 and 27.5 ± 1.1% for L:EB films but increased to 397.2 ± 21.1 and 91.5 ± 2.0% for the DL:EB counterpart. The lower glass transition temperature of DL:EB copolymers, together with the lack of crystalline domains in their structure, could be associated with the “softer” behavior of the DL:EB films in comparison to L:EB films. This same behavior was also observed at body temperature (i.e., 37 °C) but with lower strength-related values due to the proximity of the glass transition temperature and the corresponding glassy-to-rubbery transition.

Regarding the effect of CNTs, the addition of CNTs significantly increased Young’s modulus in DL:EB films from 390.3 ± 28.8 MPa for 0% CNTs to 763.9 ± 61.8 for 0.1% CNTs and 1337.5 ± 102.6 MPa for 0.5% CNTs. However, the DL:EB sample containing 1% CNTs showed a decreased Young’s modulus of 1181.4 ± 41.9 MPa, which can be ascribed to the poor dispersion of CNTs in the polymer matrix as observed in TEM images [15]. For L:EB, Young’s modulus did not experience a significant increase with respect to the number of CNTs, but there was a small difference between the samples with CNTs (2318.0 ± 187.1 MPa for 0.1% CNTs, 2395.6 ± 158.6 MPa for 0.5% CNTs and 2449.5 ± 299.2 MPa for 1% CNTs) and those without (2076.6 ± 171.0 MPa for 0% CNTs). Regarding the yield strength for DL:EB, which was calculated as offset yield strength at 10%, it suffered an increase from 0.7 ± 0.1 MPa for 0% CNTs to 1.6 ± 0.2 MPa for 0.1% CNTs and 2.5 ± 0.3 MPa for 0.5% CNTs. Contrarily, the 1% CNTs DL:EB sample showed a decrease to 1.2 ± 0.2 MPa, in line with the decrease shown in Young’s modulus. On the other hand, for L:EB, the yield point increased slightly from 14.1 ± 2.5 MPa for 0% to 16.3 ± 2.3 MPa for 0.1%, 17.2 ± 3.4 MPa for 0.5% and 18.3 ± 1.2 MPa for 1%, also in line with the trend observed for Young’s modulus. The elongation at break was also affected by the incorporation of CNTs. In the case of DL:EB, this value decreased with the addition of carbon nanotubes (397.2 ± 21.1% for 0%, 382.0 ± 25.5% for 0.1% CNTs, 322.0 ± 17.8% for 0.5% CNTs and 307.4 ± 19.1% for 1% CNTs). Similarly, for L:EB, the elongation at break decreased slightly with the addition of carbon nanotubes (177.2 ± 7.6% for 0%, 171.6 ± 13.6% for 0.1% CNTs, 165.4 ± 15.8% for 0.5% CNTs and 123.2 ± 10.3% for 1% CNTs). For strain recovery, in both cases, the observed value were not influenced by the number of CNTs. In this regard, the strain recovery for DL:EB samples was between 91.5 ± 2.0% and 96.6 ± 0.3%, while for L:EB these values were between 19.2 ± 1.8% and 28.5 ± 0.6%. Regarding the mechanical properties at body temperature (37 °C), the effect of CNTs on the mechanical properties of the DL:EB and L:EB films was negligible. The softer behavior of the copolymers at body temperature may result in the deformation of the matrix around the nanofillers, with little to no stress transfer between the two phases.

### 3.4. Contact Angle

Regarding the contact angle measurements, it seems that the incorporation of CNTs did not affect the calculated surface energy of the films (Table 5). However, a slight difference was observed between the DL:EB and L:EB films. For DL:EB films, the total surface energy $\left(\gamma_{\mathrm{total}}\right)$ was between 78.0 and 79.1 mJ/m^2^, while for L:EB films, the $\gamma_{\mathrm{total}}$ was between 71.5 and 74.2 mJ/m^2^. The lower surface energy of the L:EB samples in comparison to DL:EB samples suggests a slightly more hydrophobic nature of the L-lactide-containing copolymers. This can be ascribed to the semicrystalline nature of this copolymer, which results in a poorer interaction with water due to the more compact polymeric chains in the crystalline phase.

### 3.5. Cell Viability Assay and Immunostaining

The interaction between the synthesized non-nanostructured samples and the neural cells was studied by seeding astrocytes (C8-D1A) on the surface of the materials. After the damage to the neural system, astrocytes migrate towards the injured site and are activated due to the proinflammatory and prooxidant environment, generating the well-known glial scar and creating a physical barrier that hinders regeneration [18,25]. In this scenario, the cytocompatibility of the polymeric samples with astrocytes may directly affect the regeneration of the injured site after implantation [17]. First, the metabolic activity of cells was tested by means of the AlamarBlue assay to study the possible impact of the sample composition and presence of CNTs (Figure 4).

Our results showed that, after 24 h of incubation, the astrocytes seeded over our polymeric scaffolds containing increasing amounts of CNT presented a higher metabolic activity compared to the cells seeded directly on the tissue culture plastic. For L:EB, the metabolic activity was incremented as the number of CNTs increased (L:EB 0% CNT 113.8 ± 8.3%, L:EB 0.1% CNT 146.2 ± 13.2%, L:EB 0.5% CNT 151.2 ±9.0%, L.EB 1% CNT 158.3 ± 12.5%) (* *p* < 0.05, Dunn’s method One-way ANOVA Analysis of Variance on Ranks). These results suggested that the incorporation of CNTs on L:EB scaffolds was beneficial for astrocyte culture at short periods. As a matter of fact, the CNTs content has been reported to have a positive impact on the viability of astrocytes in vitro compared to polymeric substrates alone [11,15,16,26]. Nevertheless, after 48 h of incubation, the L:EB scaffolds showed a decrease in metabolic activity in comparison with samples at 24 h (L:EB 0% CNT 94.5 ± 9.2%, L:EB 0.1% CNT 78.5 ± 7.2%, L:EB 0.5% CNT 121.5 ± 15.7%, L:EB 1% CNT 93.4 ± 5.6%). This result can be ascribed to the crystallization (Appendix A) of samples after incubation at 37 °C for 48 h. Indeed, L:EB samples appeared to be more fragile and bent after incubation for 48 h with the culture media, which may have resulted in a substrate that compromises the cell viability.

For DL:EB, the metabolic activity was augmented in a similar manner with respect to the cells seeded on the tissue culture plastic in all of our polymeric scaffolds, irrespective of the number of CNTs. The metabolic activity of C8-D1A cells was higher than the one observed on the tissue culture plastic after 24 h (DL:EB 0% CNT 181.5 ±3.5%, DL:EB 0.1% CNT 190.7 ± 4.3%, DL:EB 0.5% CNT 176.8 ± 2.9%, DL:EB 1% CNT 181.6 ± 2.2%); and also 48 h (DL:EB 0% CNT 232.5 ± 3.8%, DL:EB 0.1% CNT 227.6 ± 10.7%, DL:EB 0.5% CNT 241.2 ± 3.1%, DL:EB 1% CNT 234.6 ± 5.7%). Our results suggested a better cell proliferation on DL:EB scaffolds independently on the CNT content with no statistical difference between the polymeric samples. Nevertheless, it needs to be considered that the activation of reactive astrogliosis also supposes a rise in metabolic activity [26], together with a morphological change from thinner to stellate conformation with bigger body size, cellular hypertrophy and thickened processes [27]. Hence, an immunostaining assay against rhodamine-phalloidin (Rd/Ph) was performed to better assess the influence of the material properties on astrocyte activation. Our results showed no activated astrocyte morphology when seeded on L:EB, but cells appeared to have a squeezed appearance which could be ascribed to the increased metabolic activity after 24 h and the decrease after 48 h due to possible cell death. Astrocytes seeded on the DL:EB samples maintained a similar thin morphology independent of the CNT content with no appearance of activated morphology (Figure 5). Thus, although future experiments should include GFAP staining and inflammatory molecular experiments to discard this possibility of astrocytic activation, the differences in cell viability assays might be ascribed to a better proliferation of the astrocytes on DL:EB substrates rather than to the possible proinflammatory activation. We can conclude that all the DL:EB scaffolds tested were cytocompatible and supported the growth of astrocytes. Our results clearly encourage the use of soft amorphous copolymers rather than non-amorphous ones in combination with CNTs for the culture of astrocytes and possibly other neural cells.

Finally, as aligned guided regeneration is a must for correct reinnervation after injury to the nervous system [18], we also studied the effect of the nanostructure on the astrocyte alignment. Astrocytes seeded on nanostructured scaffolds were able to align, following the nanograting axis (Appendix A). An aligned nanotopography has even been ascribed to modulate the reactive response of astrocytes [28], which could be beneficial for future experiments on neural culture and tissue regeneration.

## 4. Conclusions

The present work describes the synthesis of lactide and ethylene brassylate copolymers and their nanocomposites with carbon nanotubes. The mechanical properties of the resulting copolymers can be adjusted by the composition (L-lactide vs. D,L-lactide), as well as by the incorporation of carbon nanotubes. The resulting copolymers displayed a soft and elastomeric behavior at body temperature, making them interesting as scaffolds for the regeneration of soft tissues, including neural tissue. Regarding their cytocompatibility with cells from the nervous system (i.e., astrocytes), the softer mechanical behavior and the absence of crystalline domains in the formulation containing D,L-lactide seems to be more appropriate for the interaction of C8-D1A cells in terms of metabolic activity and cell morphology. Finally, the thermoplastic nature of these copolymers was exploited for the fabrication of nanostructured films capable of supporting the aligned growth of cells on their surface. Overall, the copolymers presented in this work represent a promising alternative to the more widely studied polylactide-co-caprolactone copolymers for the fabrication of polymeric devices resembling the mechanical properties of soft tissues.

## Figures and Tables

**Figure 1 polymers-14-04656-f001:**
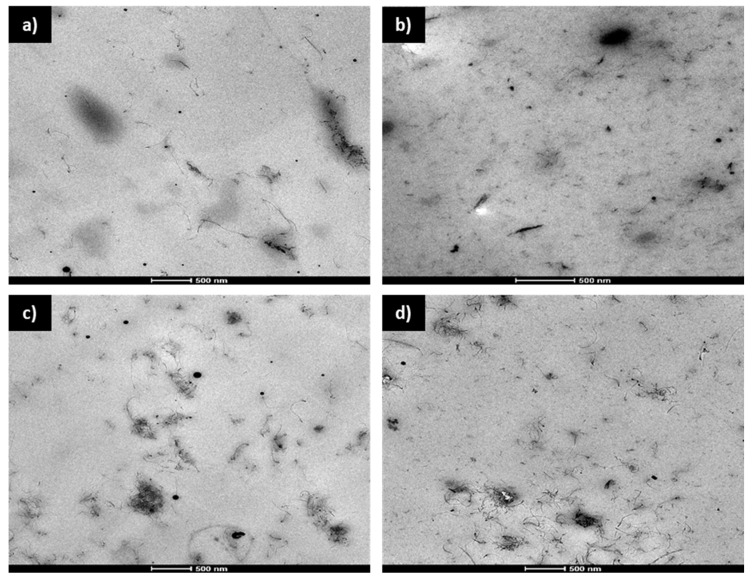
Transmission Electron Micrographs (TEM) from DL:EB samples of 0.5% CNTs (**a**) and 1% CNTs (**c**); and L:EB samples of 0.5% CNTs (**b**) and 1% CNTs (**d**).

**Figure 2 polymers-14-04656-f002:**
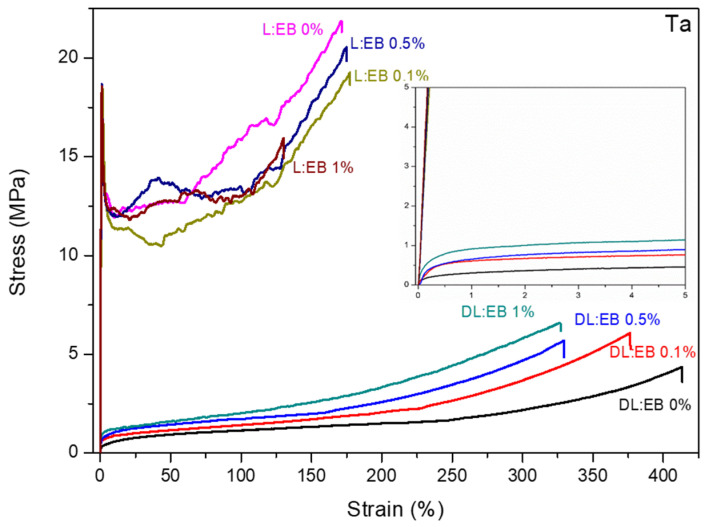
Mechanical properties of DL:EB and L:EB copolymers with different % of CNTs at ambient temperature (Ta).

**Figure 3 polymers-14-04656-f003:**
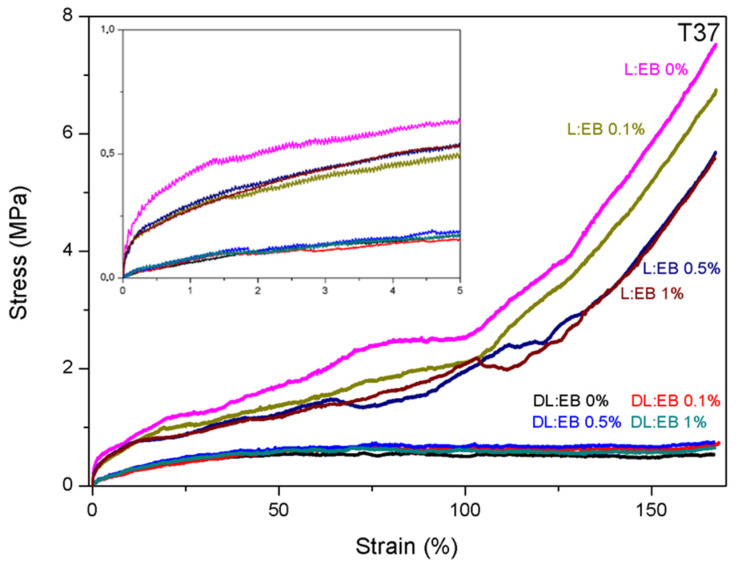
Mechanical properties of DL:EB and L:EB copolymers with different % of CNTs at body temperature (T37).

**Figure 4 polymers-14-04656-f004:**
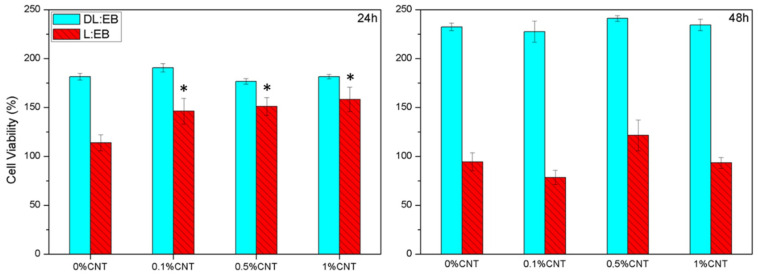
Metabolic activity of C8-D1A cells seeded on the studied samples after 24 h and 48 h. (* *p* < 0.05 compared to the sample not containing CNTs. Dunn’s method One-way ANOVA Analysis of Variance on Ranks); 100% metabolic activity was ascribed to C8-D1A cells seeded on the tissue culture plastic in the absence of our samples, and relative values were calculated for the rest of the samples.

**Figure 5 polymers-14-04656-f005:**
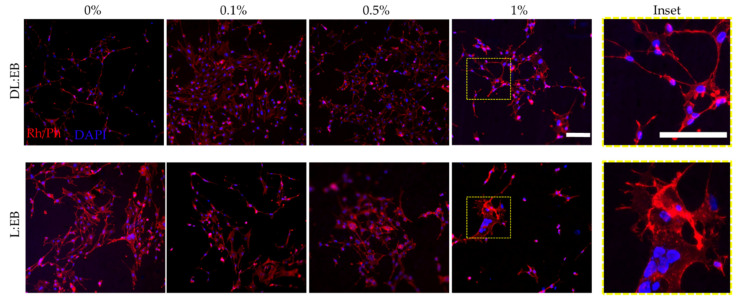
Immunostainning against rhodamine/phalloidin (Rh/Ph) and DAPI showing the morphology of the astrocytes cultured over L:EB and DL:EB containing increasing amounts of CNT after 48 h. Scale bar 100 µm in the micrographs and the insets.

**Table 1 polymers-14-04656-t001:** Characterization of the synthesized copolymers.

	DL:EB	L:EB
M_w_ (g/mol)	242,850	325,790
T_g_ (°C) (2nd scan)	33.5 ± 2.0	41.0 ± 4.5
T_m_ (°C) (1st scan)	-	158.0 ± 1.5
ΔH_m_ − ΔH_cc_ (J/g) (1st scan)	-	26.4
χ (%)	-	29.1
*l_LA_*	16.00	11.36
*l_EB_*	1.30	1.45
R	0.83	0.78
% (weight) LA	86.8	80.6
% (weight) EB	13.2	19.4
% (molar) LA	92.5	88.7
% (molar) EB	7.5	11.3

**Table 2 polymers-14-04656-t002:** Molecular weights of the resulting films after their processing and incorporation of CNTs.

M_w_ (g/mol)	DL:EB				L:EB			
%CNT	0%	0.1%	0.5%	1%	0%	0.1%	0.5%	1%
Synthesized	242,850				325,790			
Final film	207,540	190,150	194,120	215,390	250,050	210,170	256,410	216,270
ΔM_w_ (%)	14.5	21.7	20.1	11.3	23.2	35.5	21.3	33.6

**Table 3 polymers-14-04656-t003:** Thermal properties of the resulting films after their processing and incorporation of CNTs.

**Film**	**DL:EB**			
**%CNTs**	**0%**	**0.1%**	**0.5%**	**1%**
T_g_ (°C) (1st scan)	28.0 ± 2.0	25.0 ± 1.5	27.0 ± 0.5	27.0 ± 2.0
T_g_ (°C) (2nd scan)	28.0 ± 3.0	27.0 ± 3.0	29.0 ± 1.0	27.0 ± 2.0
	**L:EB**			
**%CNTs**	**0%**	**0.1%**	**0.5%**	**1%**
T_g_ (°C) (1st scan)	30.0 ± 1.0	34.0 ± 1.0	33.5 ± 1.0	37.4 ± 4.2
T_m_ (°C) (1nd scan)	156.0 ± 0.0	159.0 ± 0.0	158.0 ± 1.0	159.3 ± 0.7
ΔH_m_-ΔH_cc_ (J/g) (1nd scan)	12.0 ± 1.0	11.0 ± 0.5	8.6 ± 2.7	10.1 ± 1.6
T_g_ (°C) (2nd scan)	30.0 ± 0.0	36.0 ± 2.0	33.5 ± 2.0	34.7 ± 2.4

**Table 4 polymers-14-04656-t004:** Mechanical properties of the resulting films at ambient temperature (Ta) and body temperature (T37).

	DL:EB	Young’s Modulus (MPa)	Secant Modulus at 2% (MPa)	Ultimate Tensile Strength (MPa)	Elongation at Break (%)	Yield Strength or Offset Yield Strength at 10% (MPa)	Strain Recovery (%)
Ta	0%	390.3 ± 28.8	19.4 ± 1.7	3.8 ± 0.4	397.2 ± 21.1	0.7 ± 0.1 *	91.5 ± 2.0
T37			6.6 ± 0.6	0.7 ± 0.1	>167	0.3 ± 0.1 *	72.7 ± 2.2
Ta	0.1%	763.9 ± 61.8	26.7 ± 5.6	5.9 ± 0.2	382.0 ± 25.5	1.6 ± 0.2 *	95.6 ± 1.0
T37			4.5 ± 0.2	0.7 ± 0.1	>167	0.3 ± 0.1 *	83.2 ± 3.1
Ta	0.5%	1337.5 ± 102.6	146.4 ± 9.9	9.4 ± 0.5	322.0 ± 17.8	2.5 ± 0.3 *	93.4 ± 0.8
T37			5.7 ± 0.6	0.9 ± 0.1	>167	0.4 ± 0.1 *	74.1 ± 3.8
Ta	1%	1181.4 ± 41.9	77.5 ± 13.1	5.4 ± 0.4	307.4 ± 19.1	1.2 ± 0.2 *	96.6 ± 0.3
T37			6.3 ± 0.5	0.8 ± 0.1	>167	0.4 ± 0.0 *	71.5 ± 4.8
	**L:EB**						
Ta	0%	2076.6 ± 171.0	638.8 ± 81.2	21.6 ± 0.4	177.2 ± 7.6	14.1 ± 2.5	27.1 ± 1.1
T37			23.5 ± 1.1	6.9 ± 0.6	>167	0.9 ± 0.1 *	66.3 ± 3.4
Ta	0.1%	2318.0 ± 187.1	729.8 ± 119.1	19.1 ± 2.4	171.6 ± 13.6	16.3 ± 2.3	28.5 ± 0.6
T37			30.6 ± 0.8	6.4 ± 0.5	>167	0.8 ± 0.3 *	65.0 ± 2.5
Ta	0.5%	2395.6 ± 158.6	689.3 ± 105.8	19.1 ± 3.2	165.4 ± 15.8	17.2 ± 3.4	19.2 ± 1.8
T37			24.1 ± 2.3	6.0 ± 0.6	>167	0.8 ± 0.2 *	61.5 ± 2.9
Ta	1%	2449.5 ± 299.2	679.7 ± 78.5	18.3 ± 1.2	123.2 ± 10.3	18.3 ± 1.2	21.2 ± 0.6
T37			16.5 ± 1.5	5.8 ± 0.6	>167	0.8 ± 0.2 *	62.4 ± 3.6

The mechanical proper * Offset Yield strength was calculated at 10% of strain using the secant modulus at 2% as elastic modulus (E).

**Table 5 polymers-14-04656-t005:** Contact angle measurements with water and diiodomethane as liquid test.

%CNTs	θ_water_ (°)	θ_diiodomethane_ (°)	$\gamma_{s}^{d}$ (mJ/m^2^)	$\gamma_{s}^{p}$ (mJ/m^2^)	$\gamma_{total}$ (mJ/m^2^)
**DL:EB**					
0%	69.5 ± 5.4	25.8 ± 0.4	45.9	32.8	78.7
0.1%	71.0 ± 1.2	23.5 ± 1.0	46.7	31.3	78.0
0.5%	70.9 ± 0.7	19.9 ± 1.5	47.8	31.3	79.1
1%	71.0 ± 1.7	20.5 ± 0.6	47.6	31.3	78.9
**L:EB**					
0%	73.8 ± 2.1	29.9 ± 0.8	44.3	29.0	73.3
0.1%	75.3 ± 1.4	28.6 ± 0.7	44.8	27.5	72.3
0.5%	76.1 ± 2.6	29.0 ± 2.5	44.6	26.8	71.5
1%	73.1 ± 0.5	28.9 ± 0.7	44.7	29.5	74.2

## Data Availability

Not applicable.

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
