# Peer review of "Lactide and Ethylene Brassylate-Based Thermoplastic Elastomers and Their Nanocomposites with Carbon Nanotubes: Synthesis, Mechanical Properties and Interaction with Astrocytes"

_polymers, 2022, doi:10.3390/polym14214656_

Round 1
Reviewer 1 Report
This manuscript reports an experimental study of lactide and ethylene brassylate copolymers and their nanocomposites with carbon nanotubes. both L- and D,L-lactide were copolymerized with ethylene brassylate. Its structure, properties and elastomeric behavior are characterized using DSC, GPC, NMR, tensile test, TEM, etc. The work performs analysis with well-established methods, and the paper is written and organized well. The results are interesting for the research community, and the topic is in the scope of the journal. I can recommend publication of the paper after the following issues are addressed.
Comments:
1. Figures of the synthesized sample are preferred to show the texture and color the different samples.
2. CNTs are dispersed in the matrix from 0% ~ 1%, it is better to discuss more about the reason of the range selected.
3. It is not clear if all the tests are carried out within a few weeks or a few months after the samples are synthesized. Could the authors provide more discussion about the time-dependence of the properties of the samples?
Author Response
Dear reviewer
We highly appreciate your comments. We have addressed all of your concerns accordingly:
- Figures of the synthesized sample are preferred to show the texture and color the different samples: We have included a supplementary figure (Figure S1) showing the samples after their synthesis and processing with carbon nanotubes.
- CNTs are dispersed in the matrix from 0% ~ 1%, it is better to discuss more about the reason of the range selected: We have included a deeper explanation in section "2.3 Incorporation of MWCNT in the polymeric matrix and composite preparation": This range of carbon nanotubes was selected based on our previous work [15], where the resulting polymeric matrix showed appropriate electrical conductivity and a slight stiffening with respect to the pristine polymer at these MWCNT ratios.
- It is not clear if all the tests are carried out within a few weeks or a few months after the samples are synthesized. Could the authors provide more discussion about the time-dependence of the properties of the samples?: More detailed information about the effect of time on the mechanical properties is provided in both section "2.3 Incorporation of MWCNT in the polymeric matrix and composite preparation" and section "3.2 Incorporation of MWCNTs in the polymeric matrix and preparation of composites". In section 2.3.: The films were stored at -20 °C until their mechanical characterization to avoid any possible aging during their storage. In section 3.2.: It should be noticed that the DSC thermograms did not show any enthalpy relaxation around the Tg in the first scan, which confirms that the samples did not undergo any aging process that may affect their final mechanical properties.
Reviewer 2 Report
The paper “Lactide and ethylene brassylate-based thermoplastic elastomers and their nanocomposites with carbon nanotubes: synthesis, mechanical properties and interaction with astrocytes” studied the mechanical and practical application properties of two different kinds of thermoplastic elastomers (DL:EB and L:EB). As Tg is close to human body temperature, the tunable properties sensitive to bio-tissues make these materials potentially important values and this research itself is of high quality. I suggest publication after some minor modifications:
1. Line 116: “firstly” to “first”
2. Section 2.3, preparation “under stirring” is frequently used. Did the author mean prepared and then the solvent was stirred?
3. Line 121: “resulting product” to “resultant product”
4. Line 181: “the successful dispersion” to “dispersion quality” or “property”
5. Line 187: for tensile testing, the sample needs to be clamped so that the stretched part is shorter than the length. Please specify the dimension concerning this.
6. Figure 1: the thickness/length of CNTs were not mentioned. Even though it cannot be well defined in these low-resolution TEMs, an approximately range would be helpful.
7. Line 322 and Figure 3: the plateau of stress after strain over 50% is not really linear. Maybe does the author consider the true stress to be linear?
8. Paragraph starting from line 344: I found the listing of the parameters in words and tables hard to follow. Even though it is only 4 sets of measurements, it would be much easier to visualize to plot some of the important parameters in a figure.
9. Figure 4: what does the “*” mean in left figure?
Author Response
Dear reviewer
We highly appreciate your comments and agreed with all of them. We have accordingly addressed your main concerns point by point:
- Line 116: “firstly” to “first” : corrected in the text.
- Section 2.3, preparation “under stirring” is frequently used. Did the author mean prepared and then the solvent was stirred?: The sentence has been modified: First, the copolymer was dissolved in dichloromethane. Different amounts of MWCNT (0.1, 0.5 and 1 wt.%) were added while the polymeric solution was under stirring.
- Line 121: “resulting product” to “resultant product”: corrected.
- Line 181: “the successful dispersion” to “dispersion quality” or “property”: corrected
- Line 187: for tensile testing, the sample needs to be clamped so that the stretched part is shorter than the length. Please specify the dimension concerning this: The following information has been added to the section "2.7 Tensile test": Samples of 100 mm in length and 10 mm wide were cut from films of 250 µm average thickness. The initial distance between the clamps was fixed at 50 mm.
- Figure 1: the thickness/length of CNTs were not mentioned. Even though it cannot be well defined in these low-resolution TEMs, an approximately range would be helpful: The following information has been included in section "2.1. Materials": Carbon Nanotubes (MWCNT), having an average diameter of 10–15 nm with 5–15 walls and 1–10 μm in length, were supplied by Arkema (France).
- Line 322 and Figure 3: the plateau of stress after strain over 50% is not really linear. Maybe does the author consider the true stress to be linear?: The comment from the reviewer is highly valuable. However, measuring the true stress would require to measure "in real time" the true dimensions of the sample while stretching. Since this was not possible in our case, we decided to slightly change the sentence to clarify this concept: Figure 3 shows the elastomeric behaviour of both (co)polymers DL:EB and L:EB at body temperature (37 °C). L:EB samples showed the strain-hardening phenomena at high deformations, whereas DL:EB samples displayed a more linear behaviour, keeping a constant engineering stress value with the deformation.
- Paragraph starting from line 344: I found the listing of the parameters in words and tables hard to follow. Even though it is only 4 sets of measurements, it would be much easier to visualize to plot some of the important parameters in a figure: We completely agree with the reviewers suggestions. Therefore, we have included two supplementary figures (Figures S5 and S6) to plot the data.
- Figure 4: what does the “*” mean in left figure?: As explained in the figure caption, the "*" represents statistically significant differences with respect to the sample that does not contain CNTs.